# FEDDTPT: FEDERATED DISCRETE AND TRANSFERABLE PROMPT TUNING FOR BLACK-BOX LARGE LANGUAGE MODELS

## ABSTRACT

In recent years, large language models (LLMs) have significantly advanced the field of natural language processing (NLP). By fine-tuning LLMs with data from specific scenarios, these foundation models can better adapt to various downstream tasks. However, the fine-tuning process poses privacy leakage risks, particularly in centralized data processing scenarios. To address user privacy concerns, federated learning (FL) has been introduced to mitigate the risks associated with centralized data collection from multiple sources. Nevertheless, the privacy of LLMs themselves is equally critical, as potential malicious attacks challenge their security, an issue that has received limited attention in current research. Consequently, establishing a trusted multi-party model fine-tuning environment is essential. Additionally, the local deployment of large LLMs incurs significant storage costs and high computational demands. To address these challenges, we propose for the first time a federated discrete and transferable prompt tuning, namely FedDTPT, for black-box large language models. In the client optimization phase, we adopt a token-level discrete prompt optimization method that leverages a feedback loop based on prediction accuracy to drive gradient-free prompt optimization through the MLM API. For server optimization, we employ an attention mechanism based on semantic similarity to filter all local prompt tokens, along with an embedding distance elbow detection and DBSCAN clustering strategy to enhance the filtering process. Experimental results demonstrate that, compared to state-of-the-art methods, our approach achieves higher accuracy, reduced communication overhead, and robustness to non-iid data in a black-box setting. Moreover, the optimized prompts are transferable.

## 1 INTRODUCTION

Large language models (LLMs) have demonstrated significant success across numerous natural language processing (NLP) tasks (Brown et al., 2020; Devlin et al., 2019; Radford et al., 2019). Typically, these models are trained on a vast text corpus and then applied to various downstream tasks through fine-tuning or prompt tuning. However, task-specific data is often necessary for tuning pretrained LLMs, and this process typically relies on user-labeled data. In practice, securely leveraging these labeled data presents challenges. Data must be collected and stored for training purposes, but sharing and exchanging sensitive information can pose serious security risks and raise privacy concerns. To mitigate the risk of potential data leakage, federated learning (FL) is proposed. FL enables multiple devices to collaboratively fine-tune pre-trained LLMs on decentralized data while maintaining data privacy. Recent work, such as the bilevel optimization method (Li et al., 2024), has demonstrated efficient strategies to reduce communication overhead and improve optimization performance in FL scenarios. Additionally, federated object detection frameworks (Kim et al., 2024) and federated conditional stochastic optimization (Wu et al., 2023) have provided further insights into addressing communication and computational challenges in decentralized learning. Privacy and security remain critical in FL settings, and proactive defenses against model poisoning attacks, such as RECESS (Yan et al., 2023), help safeguard model integrity while fine-tuning LLMs in federated environments. Moreover, techniques like personalized federated learning (Yan et al., 2024) have introduced new ways to enhance the adaptability of global models to specific client data, addressing the heterogeneity often encountered in FL systems.

When applying federated learning (FL) for tuning pre-trained LLMs, existing approaches can be categorized into *federated fine-tuning* and *federated prompt tuning*. However, both methods have their limitations. For *federated fine-tuning*, the primary challenges include: (1) clients' limited access to the parameters of pre-trained language models (PLMs), (2) significant computational and storage demands on local clients, and (3) high communication overhead within the FL system. These factors make federated fine-tuning impractical in real-world scenarios. In practice, devices primarily interact with LLMs by invoking LLM APIs, which do not grant clients access to model parameters, thus preventing local training. Moreover, even if access were available, devices with limited computational resources would struggle to perform local LLM fine-tuning (Zhou et al., 2024). Several approaches have been proposed to address the challenges posed by client heterogeneity and communication costs, such as leveraging model architectures designed to improve performance in FL systems despite data heterogeneity (Pieri et al., 2023), as well as bilevel optimization methods that offer communication-efficient solutions for FL systems (Yang et al., 2024b). Additionally, methods like dynamic personalized federated learning (Panchal et al., 2022), model reassembly techniques (Wang et al., 2024), and federated multi-objective optimization frameworks (Yang et al., 2024a) offer solutions for efficient model adaptation in decentralized environments. These innovations, which target the optimization of client-specific models and data distribution challenges, may also inform strategies for fine-tuning models in decentralized contexts.

An alternative approach, *federated prompt tuning*, as proposed by `FedBPT` (Sun et al., 2023), focuses on optimizing continuous prompts injected into text while keeping the PLM parameters frozen. Although this method reduces computational costs for clients, continuous prompts still face several limitations: (1) they are model-specific and cannot be directly applied to prediction APIs, which only accept discrete inputs, (2) continuous prompts lack interpretability, and (3) they lack transferability, meaning they cannot be seamlessly applied to other LLMs. To improve communication efficiency, methods like spectral co-distillation (Chen et al., 2023) and one-pass distribution sketches (Liu et al., 2024) have been explored, targeting efficient aggregation and reduced overhead. Furthermore, the issue of communication efficiency and local model performance trade-offs has been explored in works (Li & Huang, 2024), where the tension between local client computations and global model performance is thoroughly examined, providing further insight into optimizing federated learning strategies.

To address the aforementioned challenges, we propose `FedDTPT`, On the client side, we employ a token-level discrete prompt tuning strategy. Given the absence of a probability distribution in the inference results, we implement gradient-free prompts optimization through a feedback loop based on prediction accuracy. On the server side, we utilize an attention mechanism grounded in semantic similarity to filter prompt tokens from all clients. This mechanism identifies the most representative discrete tokens. Additionally, we enhance the filtering effectiveness by employing an inflection point detection in embedding distances and a Density-Based Spatial Clustering of Applications with Noise (DBSCAN) clustering strategy. We conducted experiments on multiple datasets using SOTA PLMs. The results indicate that, in comparison to the current state-of-the-art techniques, our methodology attains superior accuracy, diminished communication expenses, and resilience to non-iid data within a black-box framework. Furthermore, the refined prompts exhibit transferability. Our contributions include:

- **Problem Novelty**: In this work, we introduce a new problem setting: discrete prompt learning in black-box federated learning. This setting enables the learning of transferable and interpretable prompts while safeguarding both the privacy of the server's model parameters and the client's data.

- **Approach Novelty**: In this work, we propose `FedDTPT`, a novel discrete prompt learning framework in black-box federated learning scenarios. `FedDTPT` utilizes the novel token-level optimization strategy to update the client prompt and a token selection method based on semantic similarity to aggregate the discrete prompt.

- **Experimental effect**: Our method achieves high accuracy and low communication overhead, and its optimized prompts exhibit transferability.

## 2 BACKGROUND & RELATED WORK

**LLMs as API Service.** Due to the significant computational demands of large language models (LLMs), an increasing number of LLMs are being deployed on servers as API services. From the *model supplier's* perspective, this approach allows them to retain proprietary control over their models, avoiding open sourcing due to commercial considerations and the risk of misuse. From the *user's perspective*, even when pre-trained LLMs are available, running them locally is often prohibitively expensive or even infeasible due to hardware constraints and the need for continuous updates (Bommasani et al., 2022). Given these advantages, deploying LLMs as API services has become a mainstream approach and is now the dominant trend.

**Federated Learning.** Federated Learning (FL) is a decentralized machine learning approach where multiple clients collaboratively train a model while keeping their data local, ensuring privacy (Konečný, 2016). For *model suppliers*, FL enables large-scale training without accessing user data, reducing liability and complying with privacy regulations like GDPR [1]. For *users*, it allows participation in model improvements while maintaining control over their data. Although FL offers privacy benefits, challenges like data heterogeneity, communication costs, and system differences remain key research areas. FL is increasingly applied to LLMs, especially in privacy-sensitive applications, making it a critical tool in privacy-preserving AI.

**Prompt Tuning.** Prompt tuning has gained considerable attention in the field of large language models (LLMs). Its goal is to search for an optimal prompt using minimal examples to guide an LLM towards generating the desired output for a specific downstream task. In NLP applications, there are two main types of prompt tuning methods: (1) continuous prompt tuning and (2) discrete prompt tuning (Liu et al., 2023). In continuous prompt tuning, a sequence of continuous vectors is appended to the input text embedding. Unlike discrete prompt, which operates at the vocabulary level, continuous prompt tuning (Li & Liang, 2021) optimizes the prompt directly in the embedding space. In contrast, discrete prompt tuning involves a sequence of discrete tokens, which remain interpretable to humans.

## 3 METHOD

### 3.1 PROBLEM FORMULATION

Prompt tuning is a widely adopted Parameter-Efficient Fine-Tuning (PEFT) method for large language models (LLMs). The prompts are optimized to adapt the model to specific downstream tasks. Discrete prompt tuning refers to the independent optimization of discrete tokens $p_n \in \mathcal{P}$ within the prompt set $\mathcal{P}$, where $n$ denotes the number of tokens in $\mathcal{P}$. This approach is more interpretable than continuous prompt tuning strategies, such as soft prompt tuning. In a federated learning context, federated discrete prompt tuning involves each client $k$, where $k \in \mathcal{K}$, transmitting their local prompts $\mathcal{P}_k = \{\mathbf{p}_k^n\}_{n=1}^N$ to a central server for a knowledge exchange based on discrete prompts. The aggregated global prompt $\mathcal{P}_F = \{\mathbf{p}_F^n\}_{n=1}^N$ is then distributed back to all clients, where it is further fine-tuned on $\mathcal{D}_k = \{(\mathbf{x}_k, \mathbf{y}_k)\}_{k=1}^K$ be a private local dataset in the $k$-th client for personalized adaptation. The objective in this federated scenario can be expressed as:

$$P_k^* = \arg\min_{P_F} \sum_{k=1}^K w_k L_k \left( f\left( P_F; D_k \right) \right), \tag{1}$$

where $n$ is the number of tokens in $\mathcal{P}$, and $K$ represents the number of clients involved. Prompt tuning based on black-box LLMs refers to the process where the large model's parameters are entirely fixed, and the prompts are treated as learnable parameters. Since the gradients of the LLM are inaccessible, gradient-free zeroth-order optimization methods are commonly used instead of traditional backpropagation techniques. Compared to standard prompt tuning, pure black-box prompt tuning is a more challenging optimization task. Since the inference result of the LLM prediction API, represented as $f(\mathcal{P}; X)$, is purely textual and does not provide a probability distribution, Eq. (2), which relies on one-hot labels, is no longer applicable. Consequently, prompt optimization is performed solely at the token level, and we accordingly use a more direct measure of accuracy as

---

[1] https://gdpr-info.eu/

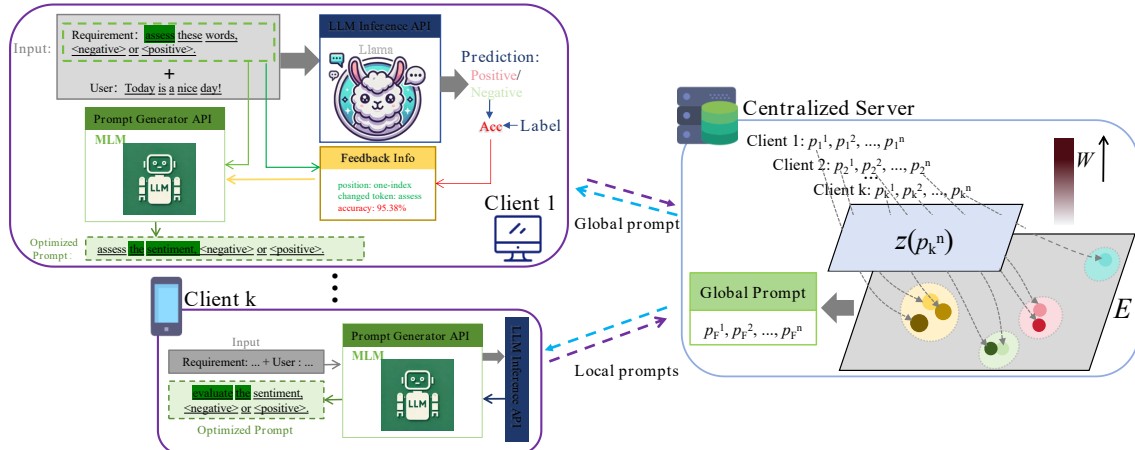

Figure 1: The structure of FedDTPT. The client uses prediction results as feedback to drive the MLM API for discrete prompt optimization. The locally optimized prompts are then uploaded to the server, where tokens are mapped to a high-dimensional latent space. Similarity calculations on these high-dimensional embeddings yield weight values $W$, and a clustering strategy is applied to select high-weight tokens. These tokens are then combined to form a global prompt, which is subsequently distributed back to the clients.

the optimization objective:

$$P_k^* = \arg \max_{P_F} \sum_{k=1}^{K} w_k A_k \left( f \left( P_F; D_k \right) \right),$$ (2)

where $A_k$ is the accuracy in client k.

## 3.2 DESIGN OVERVIEW

The overview of `FedDTPT` as shown in Figure 1. In the client optimization phase of `FedDTPT`, we adopt a token-level discrete prompt tuning strategy that establishes a new feedback mechanism for inference results to enable gradient-free prompt optimization. During the federated learning stage, we employ a semantic similarity-based attention mechanism to sample prompt tokens from all clients, selecting the most representative discrete tokens to construct optimized prompts. This approach effectively facilitates knowledge transfer across clients while preserving privacy. At the beginning of the optimization process, a public dataset $D_g$, containing representative samples, is deployed to each client to assist in computing the prediction accuracy during local prompt tuning. In each global communication round, the server first broadcasts a global prompt to all clients. In the initial round, this prompt is based on the global task and can either be carefully designed or straightforward. Subsequently, each client $k$ uses the MLM API to fine-tune the global prompt, recording the tuning information. The tuned prompt is then input into the LLM prediction API to obtain inference results and calculate accuracy. The accuracy and tuning information are aggregated as optimization feedback and fed back to the MLM API for further fine-tuning. Upon completion of local optimization, all clients upload their local prompts to the server for knowledge aggregation. The server maps the discrete tokens of all prompts to a high-dimensional latent space and employs a clustering strategy based on the secondary-range elbow judgement strategy and DBSCAN approach to cluster the embeddings. Finally, a latent space similarity-based attention mechanism is applied to sample the embeddings and generate a global prompt.

## 3.3 CLIENT PROMPT INSTRUCTION TUNING

Unlike existing black-box prompt tuning tasks, in a purely black-box setting, large language models only output prediction text without probability distributions. The absence of loss information necessitates that prompt optimization be performed solely at the token level, posing significant challenges. In the client-side optimization phase, we set accuracy improvement as the primary objec-

tive and leverage the contextual understanding capabilities of masked language models (MLMs) to achieve prompt tuning. Furthermore, we establish an inference feedback loop, which, compared to random prompt optimization using MLMs, creates a closed loop between forward inference and result feedback. This approach allows the MLM to make informed predictions based on comprehensive historical information.

Specifically, the client first receives the global prompt dispatched by the server, uses the MLM for tuning, and stores the modification details. The optimized prompt is then combined with input $x_k$ and fed into the LLM Inference API for prediction. By comparing the inference results with the labels $y_k$, we calculate the accuracy on a batch basis. Finally, in subsequent iterations, the MLM receives both the accumulated tuning modifications and accuracy results along with the prompt to be optimized. This iterative process allows the MLM to perform more informed and effective tuning. The optimization process is detailed in Algorithm 1.

---

**Algorithm 1** Token-level Prompt Optimization with Inference Feedback for Client $k$

> **Input:** Global prompt $P_{\text{global}}$, client data $\mathcal{D}_k = \{(\mathbf{x}_k, \mathbf{y}_k)\}_{k=1}^K$
> **Output:** Optimized prompt $P_k^*$ for accuracy $A_k$
> 1: Initialize $\mathcal{P}_k = \{\mathbf{p}_k^n\}_{n=1}^N \leftarrow P_{\text{global}}$
> 2: **for** iteration = 1 to max_iterations **do**
> 3:     **Optimization Objective:**
> 4:     $P_k^* = \arg\max_{P_k} A_k \left( f\left(P_k; D_k\right)\right)$
> 5:     **MLM Tuning:**
> 6:     $P_k^* \leftarrow \text{MLM API}(P_k)$
> 7:     **Inference and Accuracy Calculation:**
> 8:     predictions $\leftarrow$ LLM Inference API$(P_k^*, x_k)$
> 9:     accuracy$_k \leftarrow$ calculate_accuracy(predictions, $\mathbf{y}_k$)
> 10:    **Feedback fusion:**
> 11:    feedback_info $\leftarrow$ (modifications, accuracy$_k$)
> 12:    **Next iteration:**
> 13:    $P_k^* \leftarrow \text{MLM API}(P_k, \text{feedback\_info})$
> 14: **end for**
> 15: **return** $P_k^*$ as the optimized prompt for client $k$

---

Additionally, to address potential data imbalance during accuracy calculation in each iteration, we introduce a small, balanced public dataset to assist in accuracy computation. Specifically, during the accuracy calculation for each batch of client $k$'s data, the public dataset is incorporated as auxiliary data. This approach effectively mitigates the impact of data imbalance and helps to counteract non-iid data distribution issues.

## 3.4 SERVER PROMPT INSTRUCTION AGGREGATION

During client-side optimization, each client sends its locally optimized prompt to the server for knowledge exchange. Since clients can only access token-level information, traditional global aggregation strategies, such as simple weighted averaging, are difficult to implement. To address this, we propose an attention mechanism based on semantic similarity, combined with high-dimensional clustering methods, to effectively select and merge important tokens, thereby generating a globally optimized prompt. The detailed methodology is outlined as follows:

**Mapping Tokens to a High-Dimensional Latent Space.** Each token from the prompts generated by the clients is mapped to a high-dimensional latent space. Given the need for robust contextual understanding, leveraging the embedding layers of pre-trained language models (MLMs) like BERT or RoBERTa is well-suited for this purpose, as they can project semantically similar tokens to proximate positions in the latent space. Let $\mathbf{P}_k = \{\mathbf{p}_k^1, \mathbf{p}_k^2, \ldots, \mathbf{p}_k^N\}$ represent the sequence of discrete tokens generated by the $k$-th client, where $k \in \{1, 2, \ldots, K\}$ and $N$ denotes the number of tokens in each prompt. Each token $\mathbf{p}_k^n$ is mapped to a high-dimensional embedding through a function $z$, resulting in an embedding vector $E_k^n$. The mapping function $z$ can be formally expressed as $z : \mathbf{P}_k \rightarrow \mathbb{R}^{N \times d}, \mathbf{P}_k \mapsto \mathbf{E}_k = \{E_k^1, E_k^2, \ldots, E_k^N\}$, where $E_k^n = z(\mathbf{p}_k^n) \in \mathbb{R}^d$ is the high-dimensional embedding vector corresponding to the token $\mathbf{p}_k^n$, $d$ is the dimensionality of the latent

space, and $\mathbf{E}_k$ is the matrix of embeddings for all tokens in the $k$-th client's prompt. To incorporate context and semantics into the embeddings, the mapping function $z$ may depend on additional parameters, such as contextual weights $\theta$ from a pre-trained language model.

$$E_k^n = z(\mathbf{p}_k^n; \theta) = \text{MLM}_\theta(\mathbf{p}_k^n) \tag{3}$$

where, $\theta$ represents the parameters of the pre-trained language model (MLM), such as BERT or RoBERTa, $\text{MLM}_\theta$ denotes the model's embedding layer that captures the context and semantic similarity of each token. Therefore, the overall mapping process for all tokens from all clients can be expressed as a set:

$$\mathcal{E} = \bigcup_{k=1}^{K} \mathbf{E}_k = \bigcup_{k=1}^{K} \{E_k^1, E_k^2, \ldots, E_k^N\} = \bigcup_{k=1}^{K} \{z(\mathbf{p}_k^1; \theta), z(\mathbf{p}_k^2; \theta), \ldots, z(\mathbf{p}_k^N; \theta)\} \tag{4}$$

where $\mathcal{E}$ represents the set of all high-dimensional embeddings for tokens across all clients.

**Attention-Based Weight Calculation via Semantic Similarity.** To compute the semantic similarity between tokens, we use the cosine similarity between their high-dimensional embeddings. For a token $\mathbf{p}_k^n$ from the $k$-th client and a token $\mathbf{p}_{k'}^{n'}$ from another prompt (client $k'$), the cosine similarity is given by:

$$\text{sim}(E_k^n, E_{k'}^{n'}) = \frac{E_k^n \cdot E_{k'}^{n'}}{\|E_k^n\| \|E_{k'}^{n'}\|} \tag{5}$$

where: $E_k^n \cdot E_{k'}^{n'}$ denotes the dot product of the embeddings. $\|E_k^n\|$ and $\|E_{k'}^{n'}\|$ are the Euclidean norms (magnitudes) of the embeddings.

The attention weight $w_k^n$ for a token $\mathbf{p}_k^n$ is computed by aggregating its cosine similarities with all tokens in other clients' prompts. This can be expressed as:

$$w_k^n = \sum_{\substack{k'=1 \\ k' \neq k}}^{K} \sum_{n'=1}^{N} \text{sim}(E_k^n, E_{k'}^{n'}) \tag{6}$$

where $k'$ iterates over all clients except the $k$-th client, $n'$ iterates over all tokens in the prompt of client $k'$, and $\text{sim}(E_k^n, E_{k'}^{n'})$ is the cosine similarity between the embedding $E_k^n$ and each embedding $E_{k'}^{n'}$. To normalize the attention weights across all tokens in a prompt, we apply a softmax function to obtain a normalized weight $\alpha_k^n = \frac{\exp(w_k^n)}{\sum_{n=1}^{N} \exp(w_k^n)}$, where $\alpha_k^n$ is the normalized attention weight of the token $\mathbf{p}_k^n$. The final attention vector for all tokens in the $k$-th client's prompt is $\boldsymbol{\alpha}_k = \{\alpha_k^1, \alpha_k^2, \ldots, \alpha_k^N\}$, where $\boldsymbol{\alpha}_k$ represents the normalized attention weights for all tokens in the $k$-th client's prompt, indicating the relative importance of each token based on its semantic similarity to tokens in other prompts.

**Semantic Aggregation Using High-Dimensional Clustering.** After computing attention weights for all tokens, we employ high-dimensional clustering (e.g., k-means) to further filter semantically similar tokens. The clustering process proceeds as follows: The embeddings of all tokens serve as shown in Algorithm 2. To further enhance the flexibility of token selection, we employ a strategy based on embedding distance elbow detection and DBSCAN clustering. We calculate the distances between token embeddings and sort these distances, identifying significant changes as "elbow points" or inflection points. These points are used to determine the $\epsilon$ parameter for DBSCAN clustering. Subsequently, DBSCAN forms clusters based on the density and connectivity of the embeddings. This approach allows the number of clusters and the number of tokens within each cluster to be determined by the data itself, enabling adaptive and flexible grouping. Finally, the representative tokens from each cluster are reordered according to their original positions in the respective prompts, forming a consolidated global prompt. This step ensures that the global prompt remains semantically coherent and retains the most important information from each client.

---

**Algorithm 2** Semantic Aggregation Using High-Dimensional Clustering

---

**Input:**
1: embeddings: A list of high-dimensional embeddings for all tokens
2: attention_weights: A list of attention weights corresponding to each token embedding
3: num_clusters: The number of clusters for k-means
   **Output:**
4: cluster_representatives: A dictionary containing the representative token for each cluster
5: **Step 1: Perform High-Dimensional Clustering**
6: clusters ← KMeans(n_clusters = num_clusters).fit_predict(embeddings)
7: Initialize cluster_representatives as an empty dictionary
8: **Step 2: Find the Representative Token for Each Cluster**
9: **for** cluster_id in unique(clusters) **do**
10:     cluster_indices ← [i for i, c in enumerate(clusters) if c = cluster_id]
11:     cluster_weights ← [attention_weights[i] for i in cluster_indices]
12:     max_weight_index ← cluster_indices[argmax(cluster_weights)]
13:     cluster_representatives[cluster_id] ← embeddings[max_weight_index]
14: **end for**
15: **Return** cluster_representatives: A dictionary where each key is a cluster ID, and each value is the embedding of the representative token for that cluster

---

# 4 EVALUATION

## 4.1 EVALUATION SETUP

**Pre-trained LLMs.** In our experiments, we selected two models as backbone models: DeepSeek-V2-Lite (15B parameters) (DeepSeek-AI, 2024), and Llama-3.1-8B-Instruct (AI@Meta, 2024).

**Dataset.** We conducted experiments on seven datasets from the GLUE benchmark (Wang et al., 2019): SST-2, RTE, QNLI, MRPC, QQP, WNLI, and CoLA. Additionally, we adopted the k-shot approach for prompt training, which will be explained in detail in the following sections. Due to the consistent number of classes across datasets, we used accuracy (ACC) instead of the Matthews Correlation Coefficient (MCC) to evaluate the prediction performance for the CoLA dataset. Similarly, for QQP and MRPC, ACC was used in place of the F1 score as the evaluation metric.

**Comparison Baselines.** We evaluated our pure black-box prompt-tuning federated learning method against seven state-of-the-art (SOTA) approaches. Based on the amount of information obtained about the backbone model, we categorized these methods into **white-box** and **black-box** approaches. We define white-box LLM methods as those that have access to the full parameters of the backbone model and can obtain gradient information through backpropagation.

The **White-Box** comparison methods include the following: **FedPrompt** (Zhao et al., 2023): A SOTA method that offers communication efficiency and privacy protection by employing a prompt exchange strategy to facilitate knowledge transfer between clients in federated learning. **Open-FedLLM** (Ye et al., 2024): An open-source research library for training large language models (LLMs) in a federated learning setting. OpenFedLLM allows for various configurations through custom FL methods and LLM replacements. In this study, we used the widely adopted FedAvg algorithm to implement federated learning for the backbone model. **Manual prompt**: It refers to a manually designed prompting approach based on commonly used templates for zero-shot inference.

The **Black-Box** LLM methods do not have access to the model's parameters or gradients; they can only retrieve prediction outputs and the full probability distribution generated by the model during forward inference. These methods include the following: **FedBiOT** (Wu et al., 2024): This method compresses the original LLM into a lightweight model with similar performance, which is then distributed to each client. **FedAvg-BBT** (McMahan et al., 2017; Sun et al., 2022): A hybrid method that combines the widely used federated learning approach, FedAvg, with a black-box discrete prompt tuning method called BBT.

**Implementation & Hyperparameters .** The federated learning (FL) setup of our experiments follows the frameworks of FedPrompt and FedBPT. The FL environment consists of 10 clients, with a

100% client participation rate in each training round. Additionally, we adopted the few-shot learning paradigm commonly used in large-scale model research. Following the BDPL approach, for each dataset, we randomly sampled k instances from each class to form a new training set and sampled a different set of k instances to construct a new validation set. The new test set was composed of the original validation set. Detailed hyperparameter settings can be found in Appendix A.

## 4.2 EFFECTIVENESS RESULTS

Table 1: Effectiveness Results

| Model | Methods | SST-2 | RTE | QNLI | MRPC | QQP | WNLI | CoLA | Avg |
|-------|---------|-------|-----|------|------|-----|------|------|-----|
| | | | | | White-Box | | | | |
| | FedPrompt | 87.81 | 78.28 | 85.94 | 89.80 | 87.24 | 83.13 | 78.49 | 84.38 |
| | OpenFedLLM | 81.32 | 71.83 | 77.41 | 79.81 | 79.68 | 74.29 | 71.52 | 76.55 |
| | FedPepTAO | 85.64 | 74.02 | 79.63 | 82.77 | 82.96 | 78.41 | 73.81 | 79.61 |
| Deepseek | | | | | Black-Box | | | | |
| | Manual | 90.31 | 91.42 | 86.95 | 92.68 | 82.26 | 95.43 | 82.63 | 88.81 |
| | FedAvg-BBT | 53.12 | 50.38 | 56.25 | 59.38 | 53.75 | 53.12 | 50.75 | 53.82 |
| | Our | 97.43 | 94.86 | 94.69 | 97.88 | 95.73 | 94.72 | 91.85 | 95.33 |
| | | | | | White-Box | | | | |
| | FedPrompt | 91.63 | 82.41 | 89.91 | 95.18 | 94.24 | 84.71 | 81.52 | 88.51 |
| | OpenFedLLM | 77.08 | 76.93 | 83.72 | 86.49 | 81.85 | 77.63 | 76.94 | 80.09 |
| | FedPepTAO | 86.30 | 75.81 | 87.05 | 81.29 | 86.49 | 75.82 | 77.49 | 81.46 |
| Llama-3.1 | | | | | Black-Box | | | | |
| | Manual | 87.42 | 93.79 | 85.69 | 92.94 | 86.95 | 97.60 | 81.46 | 89.41 |
| | FedAvg-BBT | 71.88 | 46.88 | 51.32 | 56.25 | 59.38 | 56.25 | 62.50 | 57.78 |
| | Our | 95.58 | 95.03 | 93.69 | 95.52 | 92.59 | 95.90 | 87.52 | 93.69 |

We first measure the accuracy of the tuned LLMs on each downstream tasks. The accuracy of `FedDTPT` and each comparsion baseline methods are listed on Table 1. From the results, we could observe that our proposed black-box tuning significantly outperforms the other basaeline methods in almost all settings. For Deepseek, in the most challenging Black-Box scenario, our method still performs exceptionally well, achieving 95.73 accuracy, whereas competing methods like Manual Prompting score lower (82.63). For Llama-3.1, the pattern of improvement is consistent. In Black-Box results, our method scores 95.52 and 95.9, respectively, far surpassing other methods like Manual Prompting and FedAvg-BBT, with the latter scoring as low as 46.88 in the Black-Box setting. This indicates that our method excels even in scenarios with limited or no model access, making it highly adaptable and robust.

## 4.3 TRANSFERABILITY RESULTS

Table 2: The Transferability Results

| Setting | Methods | SST-2 | RTE | QNLI | MRPC | QQP | WNLI | CoLA |
|---------|---------|-------|-----|------|------|-----|------|------|
| D to L | **Manual** | 90.31 | 91.42 | 86.95 | 92.68 | 82.26 | **95.43** | 82.63 |
| | **Ours** | **96.28** | **95.43** | **92.35** | **93.84** | **92.35** | 94.29 | **86.4** |
| L to D | **Manual** | 87.42 | 93.79 | 85.69 | 92.94 | 86.95 | **97.60** | 81.46 |
| | **Ours** | **96.73** | **94.32** | **95.18** | **96.43** | **94.04** | 95.2 | **90.77** |

We now explore the transferability of our trained discrete prompt. It is important to note that continuous baseline methods cannot be applied to other large language models (LLMs) besides the one on which the prompt was trained. As a result, these continuous baseline methods inherently lack transferability. In contrast, we compare the transferability of `FedDTPT` to manual prompt baselines.

The results, shown in Table 2, demonstrate that our learned discrete prompt achieves higher accuracy across almost all benchmarks. This suggests that the prompt from `FedDTPT` can be easily transferred to other LLMs for various downstream tasks, significantly reducing the prompt learning process needed to adapt to different LLMs—a common necessity as LLMs are frequently updated. The transferability highlights the advantage of discrete prompt optimization, where the learned discrete prompt can be readily deployed across multiple LLMs.

## 4.4 OVERHEAD RESULTS

Table 3: Number of trainable parameters when adopting Llama 3.1 as the backbone model

| Method | FedPrompt | OpenFedLLM | FedPepTAO | FedAvg-BBT | Our |
|---|---|---|---|---|---|
| Trainable Params. | 614k | 81k-80B | 1796k | 500 | 150 |

The number of trainable parameters when using Llama3.1 as the PLM is presented in Table 3. From the results, we observe that `FedDTPT` requires the fewest trainable parameters among all methods. This is because, unlike continuous prompt learning methods, `FedDTPT` optimizes a discrete prompt, which theoretically requires $N \times$ fewer parameters, where $N$ is the embedding size of the LLM. The results in Table 3 further highlight the advantage of discrete prompt tuning: it requires significantly fewer tunable parameters, making it more communication-efficient.

## 4.5 ABLATION STUDIES

**Client-Level.** To evaluate the effectiveness of the improvements made during client-level optimization, including the integration of prediction feedback loops and the use of MLM-API for prompt optimization, we conducted separate tests, as shown in Table 4. Here, Client-1 represents the approach without the feedback loop and uses random token replacement for optimization, while Client-2 only omits the feedback loop. The results in Table 4 demonstrate that the proposed client-level optimization significantly outperforms both Client-1 and Client-2 across all tasks. Specifically, our approach improves accuracy by a notable margin: for SST-2, it shows an increase of 14.6% over Client-1 and 5.6% over Client-2; for RTE, it improves by 7.1% and 4.1%, respectively. This clearly indicates the effectiveness of the feedback loop and MLM-API optimizations. Additionally, the results show that removing the feedback loop (Client-2) results in a consistent drop in performance across all tasks, confirming that integrating feedback is critical for enhancing model accuracy.

Table 4: The effectiveness of our propsoed client level optimization

| Method | SST-2 | RTE | QNLI | MRPC | QQP | WNLI | CoLA |
|---|---|---|---|---|---|---|---|
| **Client-1** | 83.36 | 87.92 | 84.18 | 86.35 | 81.59 | 88.48 | 79.95 |
| **Client-2** | 90.56 | 91.29 | 87.95 | 89.44 | 87.62 | 92.07 | 83.2 |
| **Our** | **95.58** | **95.03** | **93.69** | **95.52** | **92.59** | **95.9** | **87.52** |

**Sever-Level.** We evaluate the improvements made during the server optimization phase, including attention-based token selection and token clustering strategies, with the results presented in Table 5. Server-1 represents the method where high-dimensional embeddings are aggregated using a fedavg approach. Server-2 indicates the method without the clustering strategy, while Server-3 employs a fixed number of clusters. The results in Table 5 show that the proposed server-level optimization, which includes attention-based token selection and token clustering strategies, significantly outperforms other methods across all tasks. Compared to the baseline method Server-1, our approach demonstrates considerable improvements, such as an increase of 65.51% for SST-2 and 36.97% for MRPC. In comparison to Server-2, our approach shows an increase of 1.9% in SST-2 and 1.15% in WNLI, highlighting the benefits of the clustering strategy. When compared to Server-3, our approach improves accuracy by 2.31% in SST-2 and 9.1% in CoLA, confirming that a flexible, adaptive clustering strategy enhances performance across diverse tasks.

Table 5: The effectiveness of our proposed server level optimization

| Method | SST-2 | RTE | QNLI | MRPC | QQP | WNLI | CoLA |
|--------|-------|-----|------|------|-----|------|------|
| **Sever-1** | 57.75 | 62.21 | 56.48 | 59.36 | 51.89 | 47.33 | 56.52 |
| **Sever-2** | 94.68 | 93.97 | 92.81 | 93.6 | 91.77 | 94.25 | **95.65** |
| **Sever-3** | 93.27 | 94.19 | 91.22 | 94.08 | 90.92 | 94.38 | 86.16 |
| **Our** | **95.58** | **95.03** | **93.69** | **95.52** | **92.59** | **95.9** | 87.52 |

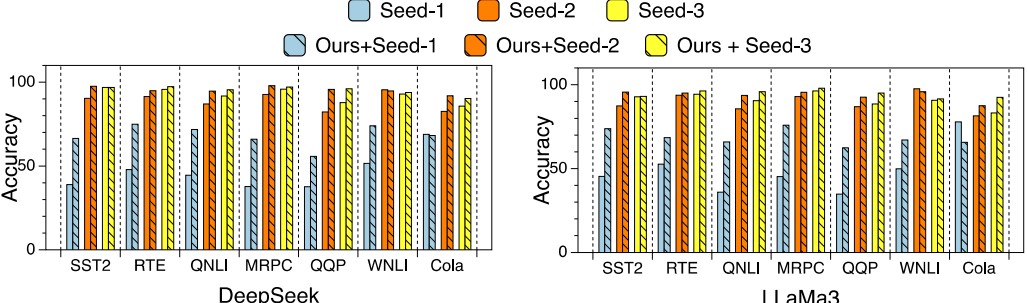

Figure 2: The accuracy of `FedDTPT` under different seed

**Seed Impact.**. To demonstrate the robustness of our optimization method against different initial prompts, we design three types of prompts—concise, moderate, and detailed formats—across each dataset and evaluate the optimization performance. The results are shown in Figure 2, with all prompt examples provided in Appendix B. In Figure 2, "Seed-n" represents the evaluation results using manual prompts directly, while "Ours+Seed-n" indicates results after applying our optimization method. For prompts of moderate and detailed formats, our approach achieves outstanding performance. Moreover, for concise prompts, although there is a larger drop in accuracy compared to other types, our method still significantly demonstrates strong optimization effects.

**Results on Non-iid Data.** To demonstrate our method's robustness against non-iid data among clients in a federated learning scenario, we conducted experiments on three datasets of varying scales: QQP, SST-2, and CoLA, as shown in Table 6. The data was simulated with Dirichlet-0.1 to model non-iid distribution. Table 6 shows that all large-model-based algorithms exhibit resistance to non-iid data, consistent with empirical observations. Furthermore, our method maintains consistently strong performance, demonstrating its superior adaptability in non-iid federated scenarios.

Table 6: Performacne of `FedDTPT` on Non-iid Data

| Benchmark | FedPrompt | OpenFedLL | FedPepTAO | Manual | FedAvg-BBT | Ours |
|-----------|-----------|-----------|-----------|--------|------------|------|
| **SST-2** | 89.27 | 76.18 | 83.21 | 88.39 | 70.73 | **94.25** |
| **QQP** | **93.61** | 80.79 | 82.92 | 86.4 | 53.62 | 91.03 |
| **CoLA** | 79.34 | 76.11 | 74.58 | 81.73 | 61.72 | **85.79** |

## 5 CONCLUSION

We propose `FedDTPT`, a FL framework that enables clients to tune discrete and transferable prompts with LLMs in black-box settings. Our approach eliminates the need for clients to access model parameters and requires only forward propagation for local training, reducing computational and storage demands for both devices and LLM service providers. Additionally, our discrete prompts are interpretable to developers and can be transferred to other LLMs without any modifications. Evaluations on several datasets using state-of-the-art PLMs show that `FedDTPT` outperforms existing white-box and black-box methods with significantly lower communication and memory overhead. Furthermore, `FedDTPT` demonstrates excellent transferability.

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
