# 6 APPENDIX A

For the few-shot setting, following the FedBPT approach, we randomly selected 100 samples (k=100) for each class in the datasets. For datasets with a large original validation set, such as QQP, following BDPL, we sampled 1,000 instances for validation.

For the federated learning setting, under Independent and Identically Distributed (IID) data conditions, we ensured uniform distribution of both the quantity and categories of data across each client. For Non-Independent and Identically Distributed (Non-IID) data conditions, we applied the Dirichlet distribution with hyperparameter $\alpha$ to control the data sparsity across different clients. The smaller the value of $\alpha$, the more imbalanced the data across clients becomes. In this study, we set $\alpha$ = 0.1.

For prompt tuning, we followed commonly used manual templates and set the number of prompt tokens to 4. The number of candidate prompt tokens was set to 100.

# 7 APPENDIX B

The three types of prompts—concise, moderate, and detailed formats—across each dataset:

SST-2:

- "judge the sentiment."
- "judge the sentiment, negative or positive ."
- "Please judge the sentiment of the User's words, your answer should be negative or positive ."

RTE:

- "Does sentence1 entail sentence2?"
- "Evaluate the relationship between these sentences. Reply with either entailment or not_entailment ."
- "Consider the logical connection between these two statements. Does the information presented in the first statement lead to the conclusion of the second statement? If so, answer entailment . If not, respond with not_entailment ."

QNLI:

- "Does the sentence entail the question?"
- "Can the information in the sentence provide an answer to the question? Respond with entailment or not_entailment ."
- "Please analyze the relationship between the question and the sentence. Does the content of the sentence provide a valid answer to the question posed? If yes, respond with entailment . If not, respond with not_entailment ."

CoLA:

- "Is the sentence grammatically acceptable?"
- "Evaluate the grammatical correctness of the following sentence. Respond with acceptable if it is grammatically correct, otherwise respond with unacceptable ."
- "Please analyze the following sentence for grammatical acceptability. If the sentence is grammatically well-formed and makes sense, answer acceptable . If the sentence is not grammatically correct or does not make sense, respond with unacceptable ."

WNLI:

- "Does the first sentence entail the second?"

- "Read the two sentences and determine whether the second sentence logically follows from the first. Respond with entailment or not_entailment ."
- "Analyze the relationship between the two sentences provided. Does the meaning of the first sentence lead to the meaning of the second sentence? If so, respond with entailment . If not, respond with not_entailment ."

QQP:

- "Are these two questions semantically the same?"
- "Compare the two questions and decide if they are asking about the same thing in different words. Answer with duplicate if they are, otherwise respond with not_duplicate ."
- "Evaluate the meaning of the following two questions. If both questions essentially ask the same thing but are worded differently, reply with duplicate . If they ask different things, reply with not_duplicate ."

MRPC:

- "Are the two sentences paraphrases of each other?"
- "Read the two sentences. Do they convey the same meaning in different words? Respond with paraphrase or not_paraphrase ."
- "Please analyze the following two sentences and determine their semantic similarity. If the sentences have the same meaning but are worded differently, answer with paraphrase . If they differ in meaning, respond with not_paraphrase ."