# OpenReview forum: "FedDTPT: Federated Discrete and Transferable Prompt Tuning for Black-Box Large Language Models"
_ICLR.cc/2025/Conference — ICLR 2025 Conference Withdrawn Submission_

### Official Review · Reviewer_XRNo · 2024-10-25

**Soundness:** 1
**Presentation:** 1
**Contribution:** 1
**Rating:** 1
**Confidence:** 3

**Summary:**

This paper proposes a federated learning method for tuning the prompts for LLM. The key idea seems to be: 1) the local clients optimize prompts by updating local prompts with local accuracy results; 2) the central server aggregates the prompts and computes the embedding of the tokens in the prompt, and uses some aggregation results to provide some updated global prompt for the local client. Some experimental results are provided to show the effectiveness of the algorithm.

**Strengths:**

1) This paper seems to provide an algorithm for federated prompt tuning.
2) Some preliminary experiment results are provided to demonstrate the effectiveness of the proposed algorithm.

**Weaknesses:**

This paper is a draft far from even being reviewed in the conference because of its writing.
1. It is questionable whether Section 3.4 is finished. 1) In the last paragraph of Page 6, it mentions some steps requiring the DBSCAN algorithm, but DBSCAN never appears in Algorithm 2. Besides, it is unclear how k-means and DBSCAN work sequentially with the text description. 2) Section 3.4 stops after introducing their algorithm, but it is unclear how the server generates the global prompt or shares global updated information with the clients in FL.
2. It is super confusing what the "MLM" is in the paper. It seems MLM is an important method/API used in the algorithm, but it is never spelled out until Page 5 despite being mentioned multiple times before that. Even within Page 5, MLM is used to refer to two different things: masked language models and (embedding layers of?) pre-trained language models. It is also mentioned that the client can use MLM API, but the input and output of such API are unclear in the paper.
3. The experiment part is also unclear and probably unfinished. 1) For example, the authors mention FedBiOT as a black-box method when introducing their baseline, but no results of this method are shown in the table. 2) The experimental results of the baseline methods seem significantly lower than the reports in other papers (e.g., results of the baseline on SST-2), which may bring doubts about whether those algorithms are implemented faithfully.

**Questions:**

NA

---

### Official Review · Reviewer_7Zdj · 2024-11-02

**Soundness:** 2
**Presentation:** 2
**Contribution:** 2
**Rating:** 3
**Confidence:** 5

**Summary:**

This paper proposes a discrete and transferable prompt tuning for black-box large language models. It adopts a token-level discrete prompt tuning strategy on the client side and an embedding-based clustering techniques on the server. Experimental results demonstrate the effectiveness of the proposed method.

**Strengths:**

1. The proposed setting is a novel problem which has not been studied before.

2. Experimental baselines include many of the recently FL-LLM frameworks, such as OpenFedLLM,FedPrompt.

**Weaknesses:**

1. The writing and presentation of the paper is often unclear and convoluted, making it difficult to follow the author's argument.
a) e.g. Eq 1's notations are inconsistent with those explained in the text, e.g. \mathcal(P) vs P, and some are left undefined, e.g. L, w. In Algo 1, it is unclear what "modifications" or "feedback_info" represents as they are not properly introduced.  It is unclear what "MLM API" represents in this paper. at one place, it refers to the "pre-trained language models like BERT", at other places such as in the figure, it appears to use "LLM API"  as MLM. Due to these representation issues, it is hard to understand from the context or figure how exactly the local optimization performs. In the experiment section, it is unclear how is "transferability" defined and how are the experiments on that performed.  In Table 2, it is unclear what is "D to L" or "L to D".

2. The technical novelty and contribution is a bit limited. Both the local optimization part and the central clustering part appear to be straightforward. If I understand correctly,  the search space for the optimized prompts are only 3 to 4 prompts (according to appendix), and it is quite obvious from the human experience to identify the best from these prompts. Therefore I am not convinced that the contribution of the proposed method is significant. It is advised that the authors provide more challenging examples on the local prompts and the resulted aggregated ones to illustrate the importance of the prompt optimization and clustering steps. Also, it is suggested that the authors discuss on the technical challenges, particularly in the FL setting with non-iid data to demonstrate the technical contribution.

3. Experimental results are missing important details, making them unconvincing. First of all, it is surprising to see that Black-boxed LLM with discrete prompts outperforms parameter fine-tuning approaches (white-boxed) across specific tasks, and by a large margin. It is unclear how are the white-boxed baselines are performed or sufficiently trained, and there are no sufficient details for the baselines. For the manual approach, it is also unclear exactly how the prompts are selected and distributed across clients. FedBiOT is listed as a baseline but no results are found.

**Questions:**

See weaknesses.

---

### Official Review · Reviewer_kvLx · 2024-11-03

**Soundness:** 3
**Presentation:** 2
**Contribution:** 2
**Rating:** 3
**Confidence:** 5

**Summary:**

The authors introduce FedDTPT, an innovative federated learning (FL) framework designed to enable clients to tune discrete, transferable prompts with large language models (LLMs) in black-box settings. This framework offers a notable advantage by eliminating the need for client access to model parameters, relying only on forward propagation for local training, thereby significantly lowering computational and storage requirements for both devices and LLM service providers. Moreover, the discrete prompts produced by this approach are interpretable for developers and can be seamlessly transferred to other LLMs without modification. Through extensive evaluations on multiple datasets with state-of-the-art pre-trained language models (PLMs), FedDTPT demonstrates superior performance over existing white-box and black-box methods, achieving substantial reductions in communication and memory overhead. Additionally, FedDTPT exhibits impressive transferability across models.

**Strengths:**

1. The authors target a more realistic setting in which the clients cannot access the embedding layer and the output distribution.

2. The authors conduct extensive experiments for evaluation.

**Weaknesses:**

1. The presentation of local optimization is not clear. How do the clients utilize the MLM to fine-tune the local prompt?

2. In FedDTPT, the clients have access to a representative public dataset. How is this dataset constructed? Having access to this public dataset makes the comparison with the other methods unfair. The authors need to demonstrate to what extent the improvement under the non-IID setting comes from the public dataset. In addition, I do not think it will be necessary to apply FL if the public dataset is "representative."

3. FedDTPT adopts an additional language model for aggregation. It seems the aggregation will involve high computational costs.

4. It is unclear how "the representative tokens from each cluster are reordered according to their original positions in the respective
prompts" during the aggregation. It looks like the algorithm cannot guarantee the selected representative tokens correspond to different positions in the respective prompts.

**Questions:**

See the weaknesses.

Minor: The authors might want to change the color of the dark blue charts in Figure 1. It is really challenging to read the context there.

---

### Official Review · Reviewer_DK9h · 2024-11-03

**Soundness:** 3
**Presentation:** 3
**Contribution:** 2
**Rating:** 5
**Confidence:** 4

**Summary:**

The paper introduces FedDTPT (Federated Discrete and Transferable Prompt Tuning), a federated learning framework designed for optimizing prompts in a black-box setting with large language models (LLMs). The framework focuses on gradient-free optimization of discrete prompts, where each client independently tunes its prompt locally, and the server aggregates these prompts to create a globally optimized prompt for iterative improvement.

**Strengths:**

The paper addresses a novel problem space, combining federated learning (FL), black-box prompt tuning, and discrete prompt optimization.

**Weaknesses:**

- Although the combination of black-box LLMs and FL is novel, prompt tuning as a standalone concept has been widely explored, even for discrete prompts. The contribution is incremental, focusing more on adapting existing methods (e.g., clustering and cosine similarity-based aggregation).
- For this paper to be distinctively valuable as a federated approach, it would need to clearly show how FedDTPT’s design solves  FL-specific challenges and why a centralized approach would not offer the same benefits. Thus far, it seems that the only way to solve statistical heterogeneity is to use a "small" public dataset. This may not be easily obtainable, which is why FL is needed in the first place.
- The results presented lack a clear demonstration of how FedDTPT performs under varying degrees of data heterogeneity. Centralized prompt learning could, theoretically, also use clustering or attention-based methods to aggregate prompts without needing an iterative FL framework.
- In the server optimization phase, choosing the highest-weighted token might bias the global prompt towards tokens that are slightly overrepresented, which may not always yield the best performance for all clients.

**Questions:**

- How does the client/server handle the convergence of prompt tuning across rounds?
- Since this is a gradient-free approach, what specific criteria or metrics are used to decide whether to retain or discard a token replacement? Is there a clear definition for convergence in the feedback loop?
- Given that the clustering process groups tokens based on similarity, how does normalization of attention weights ensure that the most meaningful tokens are selected? Is there a risk that normalization skews the representation of some clusters?
- Does sharing token embeddings with the server pose any privacy risks, particularly in cases where specific tokens might reveal sensitive information? Could these embeddings be used to infer details about the client’s local data?
- One advantage of discrete prompt tuning is interpretability. Does the final global prompt maintain this interpretability after multiple aggregation rounds, or do tokens selected for optimization risk making the prompt less clear?

---

### Note · Authors · 2024-12-27

I have read and agree with the venue's withdrawal policy on behalf of myself and my co-authors.